# Comparative Research of Internal and Border Regions: Analyzing the Differences in the Cyclical Dynamics of Industries for Industrial Policy and Territorial Development

**Galina Anatolievna Khmeleva** [1,*], **Valerii Konstantinovich Semenychev** [2],
**Anastasiya Aleksandrovna Korobetskaya** [3], **Marina Viktorovna Kurnikova** [1], **Roman Fedorenko** [4,*]
**and Balázs István Tóth** [5]

1    Institute of National and World Economy, Samara State University of Economics, 443090 Samara, Russia
2    Department of Mathematical Methods in Economics, Samara National Research University
      (Samara University), 443086 Samara, Russia
3    System Integrator "Webzavod", 443001 Samara, Russia
4    Heat and Power Department, Samara State Technical University, 443100 Samara, Russia
5    Alexandre Lamfalussy Faculty of Economics, Lamfalussy Research Centre, University of Sopron,
      H-9400 Sopron, Hungary
*    Correspondence: galina.a.khmeleva@yandex.ru (G.A.K.); fedorenko083@yandex.ru (R.F.)

**Abstract:** The differentiation in the development of regions remains a major challenge for the working out-of-state industrial and regional policies aimed at balanced and sustainable development. In theory, regional differences between internal and border regions can be explained by differences in natural resources, and economic and industrial potential, as well as by the existence of external boundaries. Border regions have higher risks in ensuring the geo-political sustainability of an industry. External boundaries, as well as differences in industry dynamics between regions, cycle stages, and industry trends, are often overlooked in industrial policy making, which in itself can be a factor of volatility. In this research based upon the Russian economy, we test the hypothesis that it is possible to define the industrial cycle with the help of the index of production. The analysis is based on the official Russian statistics from January 2005 to December 2021. To test the hypothesis, an original 12-step method of analysis was used, which allows such a mathematical model to be selected that will best describe the industry cycle and allows the trend to be estimated. The cyclic dynamics were assessed with the help of structural and parametric identification of modeling and the forecasting of trajectories of evolving dynamics based upon econophysics methodology, the use of median trends, and wavelet analysis. The comparative study was made based on the example of four sectors: the food, chemical, pharmaceutical (production of medicines and materials used for medical purposes), and automotive industries. The results show, first, that there are significant differences in the dynamics of industry cycles in both the internal and the border regions, which need to be taken into account to implement the progressive economic structure and specialization strategies of a region. Secondly, the group of border regions in the food, chemical, and pharmaceutical industries is growing at a higher rate.

**Keywords:** industry cycle; internal regions; border regions; cyclical dynamics; cycle; industrial policy; territorial development; cross-border cooperation

## 1. Introduction

Current regional development and industrial policies focus our attention on efforts to reduce regional imbalances through "smart specialization" based on job growth and wealth creation. In doing so, national well-being can only be achieved through the full realization of the economic potential of all regions (Mustafin et al. 2022). However, the question of economic inclusion cannot be fully addressed without taking into account such specific factors as the existence of borders (McCallum 1995) and the cyclicality of the economy (Sohn 2014).



Due to the popular opinion on the presence of "border effects" (McCallum 1995), the peripheral geographical position of border regions and, as a consequence, their backwardness in socio-economic development from the "mainland" part of the country in most countries (Fernandes 2019), the research of border territories in regional studies is a promising exercise for analysis aimed at the identification of specific characteristics in the development of border regions and their role in the balanced and sustainable development of a country, based on a comparative analysis of border and interior regions. Since the volatility and evolutionary nature of the economy are inherent features of a region's economic system (Tikhomirova 2006), such characteristics will allow a deeper study of the causes of the unsustainable behavior of the system and more adequate industrial policies and regional development.

In our research, the choice of criteria for the comparative analysis of border and internal regions is determined by our adhesion to L.F. Punzo's ([2001] 2015) point of view who noted that the current theory of economic development has three "wings": growth, the study of cycles, and the study of structural changes in the economy, which should be studied first. In this article, we conduct a study of the differences in the dynamics of the sectoral cycles of the border and internal regions based on the example of the Russian Federation, given the importance of cyclicality on the sustainability and balanced development of the regional economy. Sectoral regional cycles provide a better understanding of the sustainable and balanced development of a country and identify regions that contribute to overall imbalances and volatility, since at a time when a country's industrial cycle is growing, some regions may experience recession and a falling trend (Semenychev et al. 2021).

In economic studies, the essential role of cyclicality as an objective factor impacting the rate of economic growth has been proved (Geraskin and Porubova 2017; Hansen 1951). Thus, cyclical development has been shown to be a natural form of movement of the socio-economic system (Topoleva 2019). There is asynchrony caused by the overlap of different phases of cycles (Treshchevskiy et al. 2010). The concept of "cyclical vulnerability" of the Russian economy was introduced during the global economic crisis of 2008–2009, arising from the hydrocarbon-dependent national economy and openness to volatile global trade and financial flows (Smirnov 2010). Many of the works show the inertial nature of the cycles, which determines the direction and rate of change at least in the short term (Geraskin and Porubova 2017). Moreover, all other influences can only contribute to, or hinder, cyclical dynamics and the overall trend. As a result, individual (cyclical) industries are characterized by high elasticity of industry sales to GDP (Kapkaev and Kadyrov 2017). This influence necessitates cyclical investment decisions to obtain the greatest financial and material returns in the shortest possible time (Kogdenko 2019).

This article is aimed at revealing the differences in the cyclical dynamics of industries between the border and inner regions according to the scale and length of cycles in the same industries, providing information for the clarification of industrial policy and territorial development.

The article offers the following hypotheses:

**H1.** *The industry cycle can be defined by an index of the volume of production.*

**H2.** *Despite the general economic conditions, the sectoral cycles vary significantly across regions and within groups of border and interior regions.*

**H3.** *In border regions, the cyclical volatility of industries is higher than in internal regions.*

In statistical methodology, a production index is a relative measure of the change in the scale of production in the periods compared. The production index is used in the analysis of the dynamics of the physical volume of production as in the methodology Rosstat.

In Russia, 40 out of 85 Russian subjects are border regions, which account for 44 percent of the total population of Russia. The research is aimed at 82 regions with stable statistics.

For the analysis, the authors selected four sectors of the economy: the food industry, the petrochemical industry, the production of motor vehicles, trailers, and semi-trailers,

and the production of medicines and materials used for medical purposes. This choice is based on the availability of a relatively comprehensive statistical base for analysis, the level of the prevalence of these industries across Russian regions, and their importance to the economy as a whole.

The authors applied the method of modeling the industrial cycle (Semenychev et al. 2022) previously developed by the authors in the methodology of econophysics taking into account inherent mezodynamics as the object of the analysis of openness, complexity, non-linearity, non-stationarity, as well as "heavy tails" interference distribution. For trends, a linear model and 10 non-linear (parameter) models, including 8 logistic ones, were applied. The cycles were modeled by the sum of a small number of harmonics with odd frequencies, and the minimization of the action of short "bursts" for cyclicity estimation was performed by a package of 42 wavelet transformations. The nature (additive or multiplicative) of interference with trends and cycles was determined, the error of the estimates of trends from bias was reduced, and ordinal statistics ensured the robustness of the estimates of trends and cycles.

## 2. Literature Overview

### 2.1. Border Regions as an Object of Regional Analysis

In the academic literature, the solution to the question of the specific characteristics of the border region begins with the definition of this type of territory. Thus, within the framework of the territorial approach (Baklanov 2018; Bilchak and Bilchak 2018), the border region is an integral geographical territory located in close proximity to the borderline of a foreign state.

Despite the fact that the regions in the strict sense are considered the subjects of the Russian Federation in Russia, and territories of the NUTS2 level in the EU, the territorial approach to the border regions considers them to be smaller Russian border municipalities (Lazareva et al. 2020) and NUTS3-level territories located next to a border, borderlands (Svensson 2022) in EU policy and practice. However, this approach is difficult in that it is not possible to specify clearly the parameters by which the immediate border zone can be defined (Berezhnaya 2021). However, such an approach, complemented by a functional approach, turns out to be effective in the public management practice: so, cross-border functional areas, being spatially specific territorial complexes, located on two (or more) sides of a state border(s) that is not defined by administrative borders but by cross-border functional linkage (Jakubowski et al. 2022), can become an important object of management within the policy of border cooperation of a single state or neighboring states.

The second important approach to border regions is a regional approach, where specific regions become border ones, depending on the existing regional hierarchy, if they have a common borderline with foreign states (Angapova 2014).

In regional studies, the study of the specificity of border regions as a whole is connected with the identification of the economic consequences of borders between neighboring sovereign states, which can act as a barrier (Leick et al. 2021) to regional economic development, as well as a resource (Sohn 2014; Kurnikova 2021) promoting such a development. At the same time, the economic integration of neighboring countries does not offset the asymmetries of borders, expressed in price differences and differences in factor costs, for example, on the US–Mexican (Anderson and Gerber 2020), Norwegian–Swedish (Leick et al. 2021) and Chinese–Hong Kong (Chandra et al. 2022) borders.

The question of the influence of national boundaries on the development of regions has long fascinated scientists of different countries. However, not only academic interest but also policy attention to border regions generated the impetus for the development of regionalism after World War II in Europe. From that moment on, the view of the economic backwardness of the border regions (Hansen 1977) arising from the periphery of these territories, aggravated by the enclosure of space at a closed border, became generally accepted. Thus, well-known research by van Houtum (2000) showed that the economic backwardness of the border territory is manifested in the transmigration, the ageing and

the lower standard of living of the population. This thesis is supported by contemporary research: for example, Suchacek (2022) claims that during their development, border regions go from densely populated and economically important territories to sparsely populated areas, which is due to their limited infrastructure and communication links to the rest of the country.

Regional trade has traditionally been the focus of the debate around the economic impact of borders on regional development, and the research tool has been the gravitational model that was used, for example, by J. McCallum who concluded that the Canada–US regional trade patterns are 20 times lower than the ones of interregional trade in these countries (McCallum 1995), which was later adjusted to 20–50% (Anderson and van Wincoop 2003). Gravitational modelling in the analysis of interregional cross-border trade makes it possible to estimate the regional effects of easing trade restrictions between countries.

More recent research has focused on differentiating the impact of various borders on regional development. Thus, in the context of European integration, it has been shown that the more central border regions of Europe benefit more from macro integration than its external border regions (bordering non-EU countries) (Petrakos and Topaloglou 2008). B. Heider's study is notable in this respect, in which a comparative analysis of the regions of the German–Polish and German–Czech border revealed a significant positive impact of the eastern EU enlargement on the rate of population change in Germany, the Czech Republic and Poland after 2004, which, however, does not compensate for the generally weaker development of the population of border towns compared to the inland cities (Heider 2019). Measuring the supply-side border effects of European border regions, there is a stronger demand for efficiency in the use of local resources than for endowments (Capello et al. 2018).

The above-stated and other regional studies focusing on border regions often use a comparative method comparing territories over time (for example, before and after the removal of legal and administrative barriers to assess their impact on the economic growth of Europe's border regions (Camagni et al. 2019)) and space (for example, interregional studies of informality and illegality in border regions of different countries (Koff 2015)). This approach certainly allows for assessing the force of the border situation on regional development processes, using the neighboring border regions of the same country or neighboring territories on different sides of the border as objects of comparison. Such an analysis still has limitations due to the absence of the so-called "control" group for comparison, in which the border factor is completely absent (Prokop'ev and Kurilo 2016). In order to overcome this limitation, one can compare border and inland regions of the same country; this kind of analysis is widely used in the works of Russian scientists covering all regions of the Russian Federation and identifying specific features of the development of border regions on the basis of the analysis of the following indicators: GRP per capita (Starikov and Ponomareva 2018), regional budget income (Tishutina 2008), investments (Glazyrina et al. 2011) and the inflow of foreign direct investment to Russia from China (Novopashina 2012). In our research, we seek to add the knowledge of the differences between the internal and external regions to this collection, looking at the industry cycles on the basis of the Russian economy.

The territorial organization in Russia implies the division into municipal entities and their association into larger areas—regions (Russian subjects)—that differ significantly in scale. For example, the distance from the regional center of one of the Russian border regions—the city of Novosibirsk—to the border with Kazakhstan is 480 km. Therefore, Russia has already left from the "narrow" understanding of the border territory for the purposes of regional development, understanding that the reduction in negative "border effects" is only possible to consider border municipalities on a small border strip as an integral part of the economy of the region as a whole. The location of borders plays an important role in the regional economy, regardless of the distance to the industrial center of the region. If this distance is more than 100 km, the impact of the peripheral effect—

remoteness from the centre of the region, where the main centers of education, health and other services tend to be concentrated—increases. In this sense, the border regions or regions with border municipalities in their composition in Russia are in a more difficult situation than the interior regions and depend to a large extent on how simple trade and movement regimes are with neighboring territories.

It is no coincidence that in documents of strategic character, the border regions are identified as a separate category (Russian Federation 2020). Thus, in the Strategy for Spatial Development of the Russian Federation until 2025 (Russian Federation 2019), 21 subjects of the Russian Federation, located along the land border of the country, are divided into four groups depending on the contiguity with the state being a member of an international association of countries. The composition of these groups and certain administrative, territorial and socio-economic characteristics are presented in Table 1.

**Table 1.** The border geostrategic territories of Russia.

| No | Group Name | Russian Subjects | Total Area, sq. km | Population on 1 January 2022, ths. People | Average per Capita Monetary Income (per Month), rub. | Contiguous State(s) |
|---|---|---|---|---|---|---|
| 1 | Subject of the Russian Federation bordering the European Union | Leningrad Oblast | 83.9 | 1911.6 | 36,847 | Estonia, Finland |
| 2 | Subjects of the Russian Federation bordering the Eurasian Economic Union | Smolensk Oblast | 49.8 | 909.8 | 30,731 | Belarus |
| | | Altai Krai, Astrakhan Oblast, Volgograd Oblast, Kurgan Oblast, Novosibirsk Oblast, Omsk Oblast, Orenburg Oblast, Samara Oblast, Saratov Oblast, Tyumen Oblast, Chelyabinsk Oblast | 2551.6 | 23,365.3 | 30,708 | Kazakhstan |
| 3 | Subjects of the Russian Federation bordering other countries | Altai Republic | 92.9 | 221.6 | 23,798 | Mongolia, China |
| | | Republic of Tyva | 168.6 | 332.6 | 20,652 | Mongolia |
| | | Krasnodar Krai | 75.5 | 5687.3 | 43,217 | Republic of Abkhazia/Georgia |
| | | Belgorod region, Voronezh region, Kursk region, Rostov region | 210.3 | 9056.9 | 34,456 | Ukraine |
| 4 | Subjects of the Russian Federation bordering the countries of the Eurasian Economic Union as well as other countries or countries of the European Union | Pskov region | 55.4 | 613.3 | 29,332 | Belarus, Estonia, Latvia |
| | | Bryansk region | 34.9 | 1168.8 | 31,608 | Belarus, Ukraine |

It is worth noting that the Kaliningrad Oblast is not included in the list of the Russian border geostrategic territories in the Strategy for Spatial Development of the Russian Federation until 2025 (Russian Federation 2019) (e.g., in terms of the policy for spatial development it is considered to be a priority geostrategic territory, which is characterized by an exclave status); however, we include this region into the list of border territories when analyzing regional cycles.

Although balanced polycentric development is still a big problem for Russia, there are examples of successful border regions. For example, the ten most successful regions in terms of economic development in 2022 included the border regions of the Samara region



and the Tyumen region, which occupy 9th and 10th places, respectively. However, given that there are more than 20 border geostrategic territories in Russia—this is not so much.

### 2.2. Regional Cycles

Regional industries are complex systems, which according to Porter, consist of a critical mass of interconnected individual firms based on different knowledge, competencies, resources, and technologies (Porter [1980] 1998). In this sense, entrepreneurs are extremely important. As founders of new firms (Gartner 1988), entrepreneurs form a market for the supply of goods and services in the regional economy and thus create an incentive for existing firms to work better (Fritsch 2011; Porter [1980] 1998) playing a vital role in promoting regional industrial development. The role of a regulator of economic development in a region is assigned to the regional authorities that develop and implement various kinds of policies. Thus, firms and regional authorities are key players in regional industrial development.

The work of Hansen (1951) is considered to be one of the first in-depth works on industry cycles, in which the author highlighted the two-year cycle of the textile industry, explaining its features of a resource renewal technology for cotton cultivation. Hansen (1951) used similar approaches to cycles in the livestock industries. The author (Hansen 1951) defines the economic cycle through fluctuations of the most important macroeconomic variables: employment, production output, and investment.

Upshifts and downshifts in the early research of cycling have generally been associated with fluctuations in real investment. Hansen (1951) proposed distinguishing between real (capital and working assets) and financial (securities purchase) investments (1959, p. 38). Fluctuations in income, output, and employment were seen as key closely related economic characteristics of the industry. It has been noted that the volatility of investment is higher than that of consumption.

The industrial cycle is a variation of the cycle alongside the financial and commercial ones. As it refers to the production of material goods, accordingly, fluctuations are considered in relation to production volumes, prices of resources and products, employment, and investment.

In its most general form, the industrial cycle is defined as the fluctuation of actual production around its potential value (Fischer et al. 1998). Scientists have found that the share of physical assets and investment capacity of production directly affects the depth of the cycle. For example, Hansen (1951) believed that only the heavy industry was most susceptible to abrupt cyclical fluctuations, and the industry cycle was determined by increasing or decreasing purchases of goods for real investment and consumer durables.

In this article, the authors define the regional industry cycle as a dynamic process of fluctuations of economic activity within the life cycle of an industry, characterized by the repeatability of successive stages of decline and rise in the industry of a region.

The characteristics of the ups and downs of business cycles provide important information to entrepreneurs and authorities on the current state of an industry. The information allows investors to minimize the risks of investments, and the entrepreneurs, to understand what to do: increase or reduce production volumes, whether to use a new method of production, a new way of commercial use of the existing product; whether to create a new good or give it a new quality; to expand to a new market or master a new source of raw materials; or to implement organizational innovations. Accordingly, regional authorities are able to adjust regional industrial policies.

The authors share, as many scientists, the concept of the global nature of the non-linearity of mezodynamics in evolutionary economics, as set out in the fundamental monographs of Mayevsky and Kirdina-Chandler (2020), Kleiner (2021), etc.

The modern ideas about cycles in the economy are based upon the works of N. Kondratiev, K. Zhuglyar, J. M. Keynes, S. Kuznets, W. Mitchell, F. Hayek, J. Hicks, J. Schumpeter, and others. Case studies based on the example of the Russian economy examined the impact of innovation on the ups and downs in the modern economy (Glazyev 2018), as well

as the causes of crises in the Russian economy (Aganbegyan 2010; Tatarkin and Tatarkin 2010; Yakovets 2013).

Most of the known works are devoted to the study of cycles at the macroeconomic level, including the work (Semenychev et al. 2014) confirming E. Slutsky's hypothesis about the possibility of modelling cycles in the sum of small harmonics with the odd frequencies of harmonics (Slutsky 1927).

Regarding the spatially cumulative nature of growth, Myrdal (1957) suggested that leading regions are in a better position to take advantage of the opportunities created by the economic boom. It has also been found that the upswing phases of the business cycle start faster in the more developed and larger metropolises, where the agglomeration and market size create an advantage over other regions (Petrakos et al. 2005). It has been noted that during a downturn, the situation may be reversed: the more developed and metropolitan areas tend to suffer more (Petrakos and Saratsis 2000). The concept of protected regions, i.e., isolated economies that depend mainly on state transfers, is also interesting. From this perspective, protected regions do not keep pace with the rest of the aggregate economy and do not use their potential for convergence during expansion periods. They do not suffer as much as other regions during the downturns, and therefore tend to narrow their gap relative to the richer regions (Rodríguez-Pose and Fratesi 2007).

## 3. Methodology

The methods of correlation, and factor and linear regression analysis are the most common for analyzing cyclicity at the meso-level of the economy, primarily due to their presence in all known analytical packages (SPSS, Excel, Statistica).

The resulting indicators of analysis accuracy (especially forecasting) often do not exceed 50 percent level, as they are based on convenient but often inadequate assumptions about real economic practice, about the normality of laws of interference distribution and the acceptance of the strict periodicity of cycles. Nor do they consider interactions, except an additive one, in which regular decomposed components of a trajectory are considered to be independent of each other and with interference.

Low accuracy is also achieved by universal, but rather complex for real economic practice, methods: such as the apparatus of game theory and production functions, agent-oriented modeling and simulation calculations, simulation modeling, taking into account individual properties of objects of analysis; market models of imperfect competition, Markov's random processes for modelling cycles.

This article uses software and methodological tools, described in detail in the book by Semenychev et al. (2022), including mathematical models and structures, decomposition algorithms, parameters evaluation methods, as well as applied packages and functions for R program. The proposed tools and approaches are briefly explained as follows.

The tool is based on the decomposition of the trajectories of the dynamics on the trend, cycles, seasonal fluctuations, and random interference. In addition to the traditional additive and multiplicative (proportionally multiplicative) structures of interaction, mixed structures were actively used, in which part of the component interacted additively and part multiplicatively (for example, the trend is combined with cycles and multiplied by seasonality).

To ensure stability, causality, and predictability of dynamics, the simulations were performed in a parametric form: time functions for trends and ARMA models for cycles. Eleven models were used to reconstruct the trend dynamics: linear, two non-linear monotone, four logistic cumulative (S-shaped), and four logistic impulse (bell-shaped) models. The author's generalizations of logistic dynamics made it possible to describe not only symmetry (Verhulst's sigmoid) and fixed asymmetry (Gompertz's model), but also "soft" models with arbitrary adjustable asymmetry.

Such non-linear models, smoothly changing the speed and direction of dynamics in contrast to the linear model with constant growth, allow us to describe "slow" evolution on long stretches of development of economic systems. To simulate "rapid" evolution

with fast, almost instantaneous, switching trajectory to another type of dynamics due to significant internal changes in the object of analysis, the possibility of reconstruction of structural shifts was used. As a result, the diversity of trends increased from 11 to about 40 models (depending on the number of structural shifts in a particular dynamic).

The non-quadratic criterion proposed by the authors can be considered fundamentally new when reconstructing model parameters to compensate the effect of interference distributions with "heavy tails" and accidental emissions, which are characteristic of regional economic dynamics. The proposed criterion combines absolute and relative errors, taken modally. Thus, the random residue structure is addressed as both possibilities are considered.

Step-by-step procedure was used to identify parameters of non-linear models according to the proposed criterion. In the first step, the generalized simulated annealing (Xiang et al. 2013) found the local minimum region, and then with the RPROP (Igel and Huesken 2003) and NLM (Schnabel et al. 1985) algorithms approached this minimum.

It is also fundamentally new for the research to refer to the robust median approach as follows: instead of choosing one model that is the most accurate according to one or more criteria, the authors left the entire pool of reconstructed trajectories and selected the median from them at each time, including the forecast horizon. This approach effectively eliminates inadequate models that are biased against most assessments, and ensures that the most appropriate models are quickly "switched". The error connected with the selection of an adequate criterion or an "expert" model choice is also leveled.

In order to increase the stability and diversity of the trajectories from which the median is determined, bootstrapping (Efron and Tibshirani 1993) was used. In addition to the original series, models were evaluated on synthetic, mixed, and/or random residues. As a result, up to 150 models were evaluated on each row, from which the median evaluation was chosen. We should note that, in addition to the median, any percentile estimates can be used to derive interval projections (Khmeleva et al. 2021b).

The median approach was also used for cycle modelling, but another model type was used. A set of 42 wavelet transform functions collected by the authors was used to smooth the cyclic dynamics. Wavelets describe local cycles that change over time but do not allow prediction. Therefore, after smoothing and selecting median estimates, ARMA models of even orders were built. As shown in Semenychev (2004), by Z-transformation, even orders correspond to harmonic oscillations, which is consistent with E. Slutsky's theorem (Slutsky 1927).

Finally, the changeable evolutionary seasonality was distinguished by the STL method (seasonal-trend decomposition using method LOESS smoothing) (Cleveland et al. 1990).

On the basis of the above, a method of model identification of dynamics was formed, which involves the following sequence of steps.

Step 1: preprocessing of the original series of dynamics with the removal of random emissions in residues and their replacement with median smoothed values.

Step 2: determining the (additive or multiplicative) structure based on the Breusch–Pagan test and extracting the seasonal estimates using the STL algorithm.

Step 3: de-seasonalization (removal of seasonal variations from the initial series): by counting $dS_t = Y_t - \hat{S}_t$ with additive structure and $dS_t = \frac{Y_t}{1+\hat{S}_t}$ with proportional multiplicative structure.

Step 4: determining the cyclic oscillation (additive or multiplicative) structure based on the Breusch–Pagan test.

Step 5: identifying a linear trend without structural shifts $\hat{T}_t^{lin}$ on the cleaned data $dS_t$.

Step 6: de-trending with an additive structure $dT_t = dS_t - T_t$ and a proportionally multiplicative structure $dT_t = \frac{dS_t}{T_t} - 1$.

Step 7: identifying cycle estimates $\hat{C}_t$ on cleaned data $dT_t$.

Step 8: removing cyclical fluctuations from the cleaned data with additive structure $dC_t = dS_t - \hat{C}_t$ and a proportionally multiplicative structure $dC_t = \frac{dS_t}{1+\hat{C}_t}$.

Step 9: constructing the median trend without structural shifts $\hat{T}_t^{me}$. To do this, all trend models are identified, and at each point in the time series, the median value of all the trend estimates obtained is taken.

Step 10: repeating steps 6–8 on the new trend estimates $\hat{T}_t^{me}$.

Step 11: the construction of the median trend with structural shifts $\hat{T}_t^{me.sc}$ is carried out in the same way as step 9, but each trend is built both on the full data set and with the division of the series of dynamics into subsamples at the points of structural shifts.

Step 12: repeat steps 6–8 on the new trend estimates $\hat{T}_t^{me.sc}$.

Thus, in the course of the iterative procedure, we obtain a robust decomposition into parametric models of trends, cycles, and seasonality.

The statistical base of the study was made up of publicly available data of the Russian Federal Service of State Statistics, and a monthly measured index of industrial production was used.

Of the 82 regions, only regions with stable data were included in the analysis base, thus excluding regions where the industry is not developed. Therefore, different number of regions were analyzed for different industries.

Data period for the research was January 2005–December 2021 because it is from this period that official statistical bodies of Russia provided operational monthly data of the physical index of industrial production. This allowed the analysis to cover as many cycles as possible in the regions during this period and analyze the impact of global events (global crisis in 2008, sanctions against Russia in 2014).

## 4. Results

Our previous studies found out that Russian regions show different dynamics and cycle profiles in the same industries, although they are in the same macroeconomic conditions, and have similar characteristics in terms of labor and natural–climatic potential (Khmeleva et al. 2021a; Semenychev et al. 2020). For example, cyclical growth in the automotive industry in some regions may be accompanied by decline in other regions. For example, the production of motor vehicles, trailers, and semi-trailers IN the Republic of Bashkortostan showed cyclical growth by 2020 and a decline in the Samara region (Figure 1).

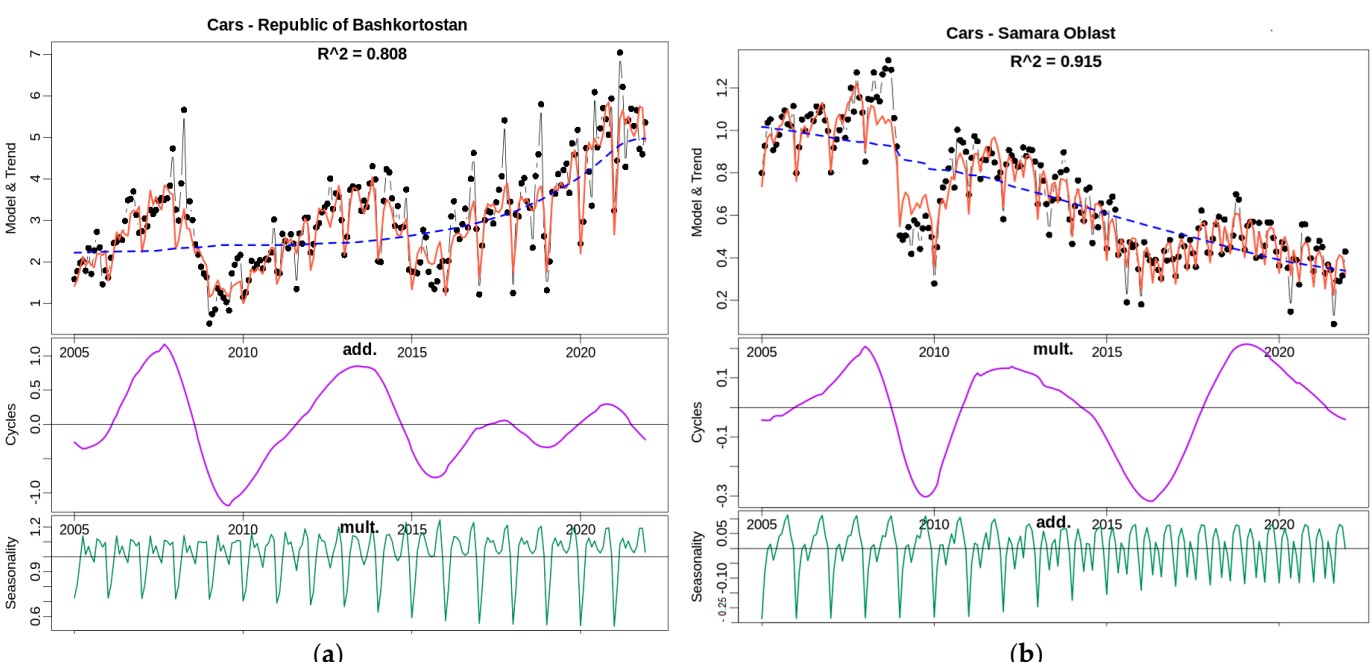

**Figure 1.** Production of motor vehicles, trailers, and semi-trailers in the Republic of Bashkortostan (**a**) and the Samara region (**b**).

The same is true for industries with relatively uniform distribution across regions and a constant demand, for example, the food industry (Figure 2). There is a falling trend of the food industry in the Republic of Altai, while, conversely, it is in growth in the Belgorod region.

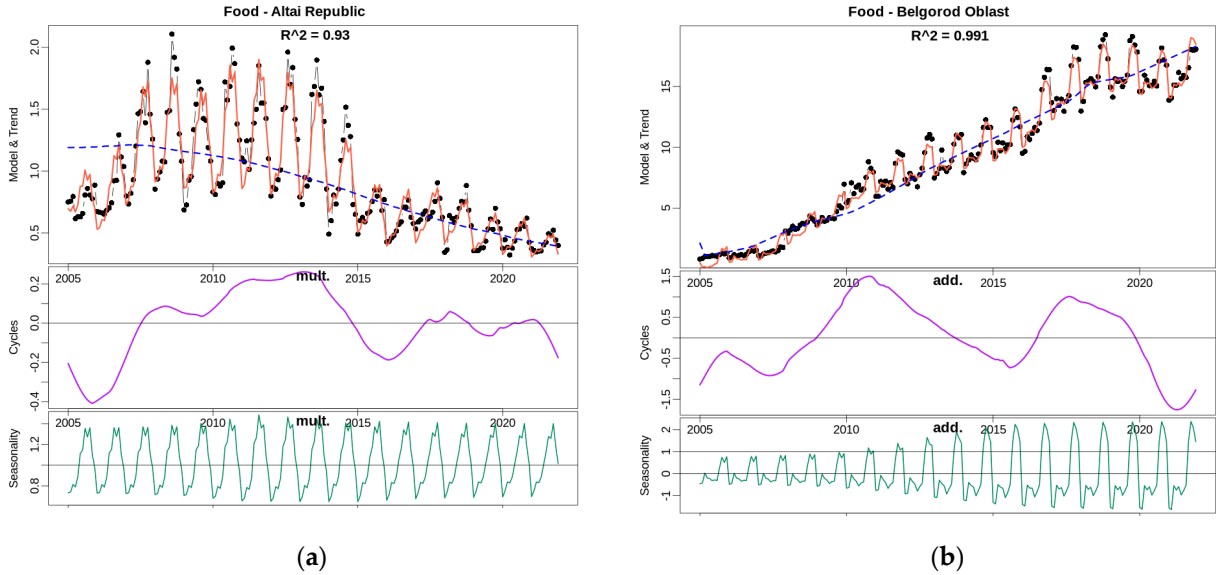

(**a**)                    (**b**)

**Figure 2.** Food industry in the Republic of Altai (**a**) and the Belgorod Oblast (**b**).

The chemical industry is also widely represented in the Russian Federation, in its 36 internal and 22 border regions, where both growth and cyclical decline can be observed at the same time (Figure 3).

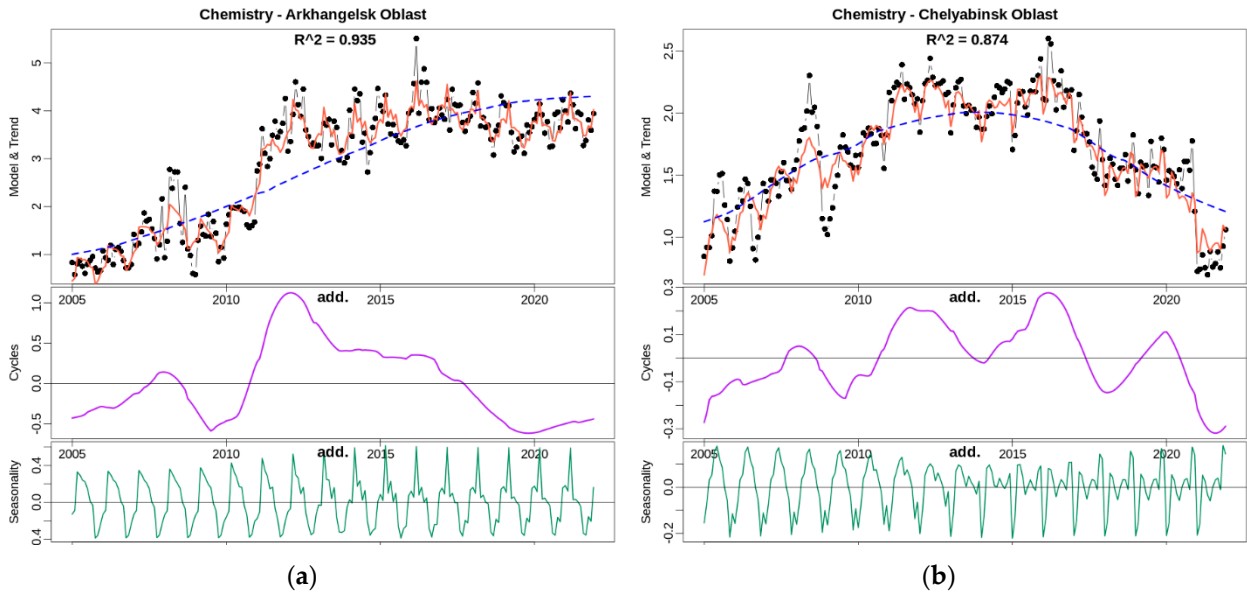

(**a**)                    (**b**)

**Figure 3.** Chemistry in the Arkhangelsk Oblast (**a**) and the Chelyabinsk Oblast (**b**).

The production of medicines and materials used for medical purposes is concentrated mainly within the country, is represented in 30% of the country's regions, and is far from uniform (Figure 4).

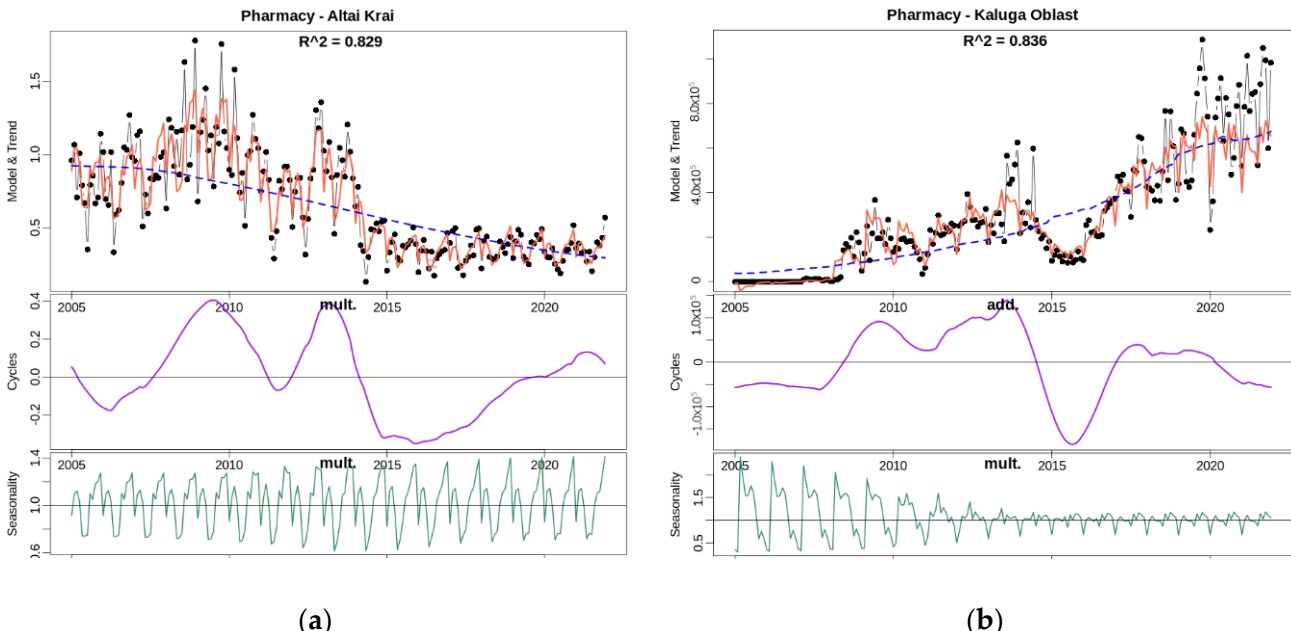

(**a**)  (**b**)

**Figure 4.** Production of medicines and materials used for medical purposes in the Altai Krai (**a**) and the Kaluga Oblast (**b**).

In the Altai Krai (a), there is a falling trend, while in the Kaluga Oblast (b), the trend is growing.

We came to an important conclusion: the size of an industry and macroeconomic conditions do not necessarily determine the situation in this industry in a region. Arguably, the internal conditions in a region are more important, dealing with how well regional business is coping with external and internal challenges.

However, there are global shocks to which individual industries have similar responses in many regions. The global financial crisis of 2008–2011 represented such a shock during the period under review. During the financial crisis, for example, there was a cyclical decline in many regions of the chemical industry and in the production of motor vehicles, trailers, and semi-trailers. During the COVID-19 period, in 2020 the regional situation was different, as can be seen in Figures 1–4. We also noticed an important point: the dynamic of the cycle is often preemptive before the influence of the big event begins. Thus, the financial crisis exacerbated cyclical decline or stagnation in some regions that began before the acute phase of the financial crisis.

It should be noted that the food industry and the production of medicinal products and materials used for medical purposes are, by their very nature, less sensitive to global shocks and, conversely, the industry cycle may shift to a growth phase during these periods.

However, we decided to go a little further and see how deep the differences between the internal regions and the regions with external borders are.

This analysis was carried out on the example of the food industry, petrochemical industry (chemistry), production of motor vehicles, trailers, and semi-trailers (automotive industry), and the production of medicines and materials used for medical purposes.

### 4.1. Food Industry

The food industry in Russia, as in any other country of the world, is the backbone of the economy. In Russia, it employs 2.7% of the working population, the share of GDP in 2021 was 2.1%, and the share of tax revenues amounted to 5.6% (Baymukhametova et al. 2023). In 2021, its production reached 8.54 billion rubles accounting for 13.56% of the processing production in total according to Rosstat.

Food production is relatively evenly distributed throughout the country. The number of regions analysed was 36 internal and 37 border regions. The figures for the cyclical evolution of the food industry in the border and internal regions are shown in Tables 2–4.

**Table 2.** Differentiation indicators of food industry cycles in the Russian border regions.

| Indicators | Trend, Growth, % | Trend, Scope, % | Average Cycle Length, Months | Seasonal Scale, % |
|---|---|---|---|---|
| Minimum | −51.74 (Jewish Autonomous Region) | 2.53 (Kaliningrad Oblast) | 35.0 (Astrakhan Oblast) | 5.16 (Kurgan Oblast) |
| Maximum | 16,834.89 (Republic of North Ossetia-Alania) | 16,834. 89 (Republic of North Ossetia-Alania) | 104.3 (Chukotka Autonomous Okrug) | 6339.74 (Chukotka Autonomous Okrug) |
| Average | 719.25 | 743.09 | 60.6 | 304.52 |
| Standard deviation | 2787.35 | 2782.69 | 17.8 | 1055.66 |
| Median | 40.46 | 71.18 | 54.6 | 86.09 |

Seasonality is variable in most regions (86%).

**Table 3.** Differentiation indicators of food industry cycles in the Russian internal regions.

| Indicators | Trend, Growth, % | Trend, Scope, % | Average Cycle Length, Months | Seasonal Scale, % |
|---|---|---|---|---|
| Minimum | −76.0 (St. Petersburg) | 4.05 (Irkutsk Oblast) | 39.6 (Stavropol Krai) | 3.38 (Komi Republic) |
| Maximum | 593.01 (Moscow Oblast) | 593.01 (Moscow Oblast) | 104.3 (Kemerovo Oblast) | 333.29 (Moscow Oblast) |
| Average | 79.44 | 91.95 | 64.2 | 90.62 |
| Standard deviation | 129.66 | 121.25 | 16.6 | 81.81 |
| Median | 48.62 | 54.63 | 63.8 | 103.84 |

Seasonality is variable in most regions (83.3%).

**Table 4.** Number of regions by cycle stages by 2021 in the food industry.

| Cycle Stage at the End (December 2021) | Number of Regions | | | |
|---|---|---|---|---|
| | Internal Regions | | Border Regions | |
| | Number | Percentage | Number | Percentage |
| G+ growth above 0 | 8 | 22 | 11 | 30 |
| D+ drop above 0 | 12 | 33 | 11 | 30 |
| G− growth below 0 | 7 | 19 | 9 | 24 |
| D− decline below 0 | 9 | 25 | 6 | 16 |
| Total | 36 | 100 | 37 | 100 |

The food industry in the border regions is more cyclical than in the internal ones. The four regions demonstrate an outstanding result—a trend growth of more than a thousand percent. Among the internal regions, the results are more moderate. In the border regions, the cycles are somewhat shorter in time, and they demonstrate a more diverse seasonality, which is probably due to the peculiarities of the natural and climatic conditions of Russia.

### 4.2. Production of Chemicals and Chemical Products

The production of chemicals and chemical products includes a wide range of goods. These include products such as plastics and synthetic rubber, fertilizers, the production of liquefied and compressed inorganic gases for industrial or medical purposes, and much

more. In 2021, this group produced goods totaling 5.26 billion rubles accounting for 8.4% of the total manufactured goods shipped.

There are many chemical industry enterprises in the Russian regions, both in the internal and border regions. We covered 22 border regions and 36 internal regions for analysis.

The cyclical differentiation in the production of chemicals and chemical products from the border and internal regions is shown in Tables 5–7.

**Table 5.** Differentiation indicators of chemistry cycles in the Russian border regions.

| Indicators | Trend, Growth, % | Trend, Scope, % | Average Cycle Length, Months | Seasonal Scale, % |
|---|---|---|---|---|
| Minimum | −83.51 (Kurgan Oblast) | 7.53 (Novosibirsk Oblast) | 41 (Bryansk Oblast) | 3.55 (Volgograd Oblast) |
| Maximum | 4004.7 (Primorsky Krai) | 4006.35 (Primorsky Krai) | 144.5 (Voronezh Oblast) | 1202.09 (Primorsky Krai) |
| Average | 444.93 | 471.89 | 63.6 | 219.9 |
| Standard deviation | 914.14 | 905.8 | 22 | 311.2 |
| Median | 58.41 | 80.33 | 57.6 | 85.23 |

**Table 6.** Differentiation indicators of chemistry cycles in the Russian internal regions.

| Indicators | Trend, Growth, % | Trend, Scope, % | Average Cycle Length, Months | Seasonal Scale, % |
|---|---|---|---|---|
| Minimum | −24.48 (Irkutsk Oblast) | 5.2 (Kemerovo Oblast) | 36.6 (Republic of Mordovia) | 2.74 (Perm Krai) |
| Maximum | 819.92 (Vladimir oblast) | 1003.96 (Khanty-Mansiysk Autonomous Region—Ugra) | 102.7 (Arkhangelsk oblast) | 553.03 (Vladimir oblast) |
| Average | 171.33 | 218.49 | 57.6 | 114.1 |
| Standard deviation | 203.51 | 240.18 | 14.5 | 133.65 |
| Median | 104.1 | 126.35 | 54.1 | 77.15 |

Seasonality is variable in all regions.

**Table 7.** Number of regions by cycle stages by 2021 in the chemical industry.

| Cycle Stage at the End (December 2021) | Number of Regions | | | |
|---|---|---|---|---|
| | Internal Regions | | Border Regions | |
| | Number | Percentage | Number | Percentage |
| G+ growth above 0 | 10 | 28 | 7 | 32 |
| D+ drop above 0 | 15 | 42 | 7 | 32 |
| G− growth below 0 | 2 | 6 | 4 | 18 |
| D− decline below 0 | 9 | 25 | 4 | 11 |
| Total | 36 | 100 | 22 | 100 |

The production of chemicals and chemical products also varies between the internal and border regions.

### 4.3. Production of Medicines and Materials for Medical Purposes

The production of medicines and materials for medical purposes is mainly concentrated in the country, with 21 internal and 6 border regions. In 2021, the amount of goods produced for this type of activity accounted for 1.29 billion rubles, which is 2% of the total manufacturing output.

The differentiation indicators of cyclical trends in the production of medicines and materials used for medical purposes in the border and internal regions are presented in Tables 8–10.

**Table 8.** Differentiation indicators of the cycles in the production of medicines and materials used for medical purposes in the Russian border regions.

| Indicators | Trend, Growth, % | Trend, Scope, % | Average Cycle Length, Months | Seasonal Scale, % |
|---|---|---|---|---|
| Minimum | −47.54 (Astrakhan Oblast) | 29.91 (Kurgan Oblast) | 56.9 (Saratov Oblast) | 3.00 (Astrakhan Oblast) |
| Maximum | 6915.61 (Samara Oblast) | 7114.67 (Samara Oblast) | 72.2 (Tyumen Oblast) | 3661.17 (Samara Oblast) |
| Average | 1919.28 | 2026.28 | 65.1 | 956.84 |
| Standard deviation | 2678.97 | 2737.91 | 6.2 | 1393.00 |
| Median | 1092.99 | 1106.43 | 65.9 | 447.93 |

Seasonality is variable in all regions.

**Table 9.** Differentiation indicators of the cycles in the production of medicines and materials used for medical purposes in the Russian internal regions.

| Indicators | Trend, Growth, % | Trend, Scope, % | Average Cycle Length, Months | Seasonal Scale, % |
|---|---|---|---|---|
| Minimum | Trend, growth, % | Trend, scope, % | Average cycle length, months | Seasonal scale, % |
| Maximum | −168.85 (Penza Oblast) | 49.39 (Tver Oblast) | 41.5 (Stavropolsky Krai) | 14.45 (Krasnoyarsky Krai) |
| Average | 3956.03 (Irkutsk Oblast) | 59,644.00 (Republic of Bashkortostan) | 80.0 (Moscow Oblast) | 107,330.74 (Republic of Bashkortostan) |
| Standard deviation | 512.6 | 3311.98 | 58.4 | 5313.69 |
| Median | 1096.18 | 12,942.67 | 10.9 | 23,376.72 |

The seasonality in all regions is variable, with the exception of the Altai Krai.

**Table 10.** Number of regions by cycle stages by 2021 in the production of medicines and materials used for medical purposes.

| Cycle Stage at the End (December 2021) | Number of Regions | | | |
|---|---|---|---|---|
| | Internal Regions | | Border Regions | |
| | Number | Percentage | Number | Percentage |
| G+ growth above 0 | 2 | 9.5 | 3 | 50.0 |
| D+ drop above 0 | 11 | 52.4 | 3 | 50.0 |
| G− growth below 0 | 3 | 14.3 | 0 | 0 |
| D− decline below 0 | 5 | 23.8 | 0 | 0 |
| Total | 21 | 100 | 6 | 100 |

The comparison of the border and internal regions shows that the border regions showed a more dynamic growth at the end of 2021 than the internal regions, where most regions experienced cyclical decline.

When comparing the cyclicality of the border and internal regions in the production of medicines and materials used for medical purposes, it is possible to observe strong inter-group and intra-group differences. The development trend of the pharmaceutical sector in the border regions is higher than in the internal regions, as it is higher in minimal (−47.54%) and maximum (6915.61%) values of the growth of the trend.

### 4.4. Production of Motor Vehicles, Trailers, and Semi-Trailers

The production of motor vehicles, trailers, and semi-trailers is found in 21 internal regions and 16 border regions in Russia. In 2021, the volume of production for this type of activity amounted to 3.23 billion rubles, accounting for 5% of total manufacturing output.

The differentiation in the cyclical production of motor vehicles, trailers, and semi-trailers from the border and internal regions is shown in Tables 11–13.

**Table 11.** Differentiation indicators of the cycles in the production of motor vehicles, trailers, and semi-trailers in the Russian border regions.

| Indicators | Trend, Growth, % | Trend, Scope, % | Average Cycle Length, Months | Seasonal Scale, % |
|---|---|---|---|---|
| Minimum | −1359.15 (Orenburg Oblast) | 28.23 (Chelyabinsk oblast) | 37.8 (Bryansk oblast) | 3.9 (Kursk oblast) |
| Maximum | 5202.26 (Oryol Oblast) | 5220.55 (Rostov Oblast) | 119.3 (Rostov Oblast) | 2393.18 (Oryol Oblast) |
| Average | 510.06 | 812.64 | 58.9 | 424.71 |
| Standard deviation | 1812.97 | 1706.87 | 19.1 | 817.79 |
| Median | −38.01 | 91.98 | 53.1 | 53.92 |

Seasonality is variable in most regions (93 percent).

**Table 12.** Differentiation indicators of the cycles in the production of motor vehicles, trailers, and semi-trailers in the Russian internal regions.

| Indicators | Trend, Growth, % | Trend, Scope, % | Average Cycle Length, Months | Seasonal Scale, % |
|---|---|---|---|---|
| Minimum | −107.57 (Tver Oblast) | 18.56 (Nizhny Novgorod Oblast) | 39.9 (Chuvash Republic) | 4.96 (Yaroslavl Oblast) |
| Maximum | 2,898,254.14 (St. Petersburg) | 2,898,254.14 (St. Petersburg) | 94.7 (St. Petersburg) | 114,516.84 (St. Petersburg) |
| Average | 126,458.65 | 126,609.14 | 60.6 | 5208.19 |
| Standard deviation | 604,213.31 | 604,198.5 | 13.6 | 23,833.64 |
| Median | −18.53 | 107.57 | 62.5 | 88.77 |

Seasonality is variable in most regions (90.4 percent).

**Table 13.** Number of regions by cycle stages by 2021 in the production of motor vehicles, trailers, and semi-trailers.

| Cycle Stage at the End (December 2021) | Number of Regions | | | |
|---|---|---|---|---|
| | Internal Regions | | Border Regions | |
| | Number | Percentage | Number | Percentage |
| G+ growth above 0 | 4 | 17 | 0 | 0 |
| D+ drop above 0 | 9 | 39 | 6 | 43 |
| G− growth below 0 | 3 | 13 | 2 | 14 |
| D− decline below 0 | 7 | 31 | 6 | 43 |
| Total | 23 | 100 | 14 | 100 |

In the production of motor vehicles, trailers, and semi-trailers, cyclical volatility is more prevalent in the internal regions. As for the border regions, cycles are shorter, and seasonality is lower.

### 5. Discussion

In this research, the models of cycles for certain industries were constructed to reveal differences in the border and internal regions in terms of cyclical industrial dynamics.

The discussion focuses on the following points.

While estimating cyclicality, the authors left only those regions with stable statistics. The application of the original 12-step method of analysis allowed the selection of a mathematical model that best describes an industry cycle compared to other models. This led to some results, which will be described below and discussed.

First, the results of this study once again confirm the significant role of cyclicality as a factor influencing economic growth (Geraskin and Porubova 2017), as can be seen from the nature of the cycle trend. Secondly, there is the phenomenon of asynchrony both between the groups of internal and border regions and within each group (Treshchevskiy et al. 2010). Third, contrary to popular belief about the economic backwardness of the border regions (Hansen 1977), they often show a higher dynamics of an industry than the internal regions. Currently border regions are increasingly becoming more successful in economic development than before. In our case, the regions with high growth in the sectors under consideration are more abundant among border regions.

Two of the three hypotheses were confirmed. As a proof of the first hypothesis (H1), it is shown that the volume of production index provided monthly by official statistics can well describe cycle dynamics using the original 12 step-by-step analysis method, which allowed the selection of a mathematical model that best describes the industry cycle compared to other models. As a proof of the second hypothesis (H2), it is shown that, despite the general conditions of economic activity, industry cycles differ significantly across regions and within groups of border and internal regions. In the third hypothesis (H3), there is no clear evidence that the volatility of the border regions is higher than in the internal ones. In any case, the standard deviation indicator in the border regions showed values higher than in the interior regions only in the food and chemical industries.

We generalized the results of the industry cyclicality in the Russian border and interior regions.

The food industry is found in almost all the analyzed regions, 36 internal and 37 border regions of Russia. The demand for food products is relatively stable, depending on the number of people and the amount of domestic and foreign exports from the region.

In the food industry, the border is very important. It is in the production of food products that regions make the best use of the potential of the border situation, although they supply products not only to neighboring countries but also much further.

Many Russian border regions are active exporters, with a steadily growing trend in the food industry, as in the Altai Krai and the Belgorod Oblast (Figure 5).

However, the argument that the border regions are exploiting the potential of the border situation only partly explains the growing trend. The second important factor is the launch of new production, as happened in the Republic of Ingushetia, which, practically, does not supply food for export.

Chemical production is found in 36 domestic and 22 border regions. Cyclical volatility is higher in border regions, as shown by the standard deviation. In four regions, the trend increased by more than 1000 percent, while it did not happen in the internal regions. The average growth rate of the border regions is four times higher than that of the internal regions. The cycle length in the chemical industry is on average higher in border regions, as well as seasonality. Border regions generally benefit from external borders. However, not all of them rely on neighboring countries as their main export destination.

The production of medicines and materials used for medical purposes is concentrated mainly in the internal regions of the country. At the same time, the border regions more successfully use the industrial base to increase production outputs. This can be judged by the indicators of the trend and median. In internal regions, the trend differentiation is higher, stronger than the difference in cycle length, which is expressed by higher standard deviation values.

The automotive industry is found in 23 interior and 14 border regions. There is also a high variation in the industry cycle between the internal and the border regions, but the internal regions have shown themselves to be more stable in growth. The reason may be the low importance of reaching the external borders and the focus of companies mainly being on domestic consumers.

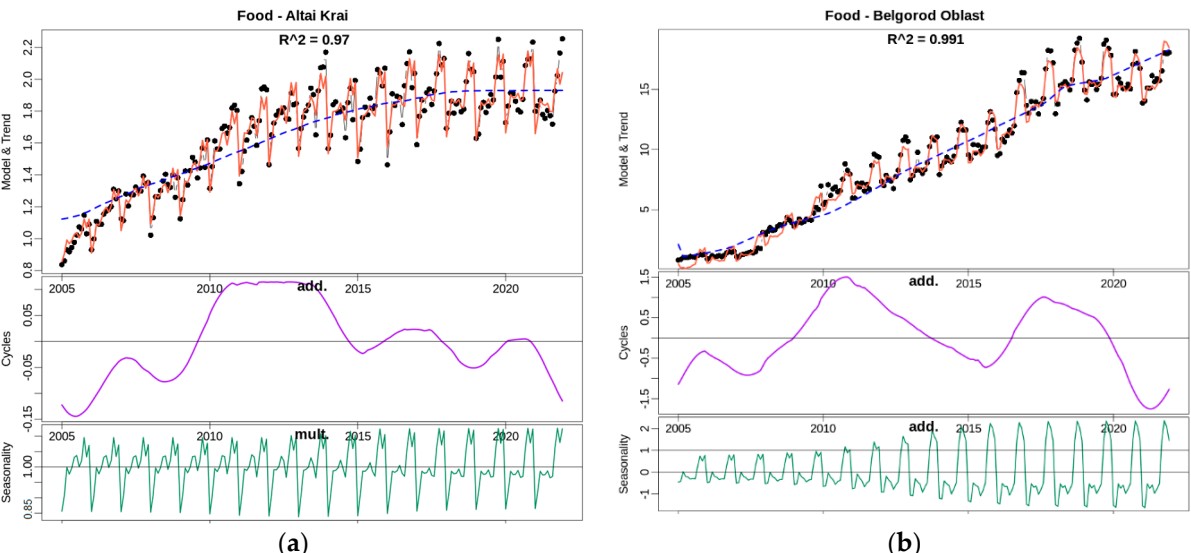

**Figure 5.** Cyclical dynamics in the food industry of the Altai Krai (**a**) and the Belgorod Oblast (**b**).

## 6. Conclusions

This research has theoretical, managerial and research implications. Thus, this work has contributed to the new knowledge on the methodology for assessing the sectoral cyclicality, and differentiation of border and internal regions, as well as on the role of territorial development policy and industrial policy. From a methodological point of view, it has broadened the idea that the industry cycle is well estimated by the index of production, since this indicator is characterized by changes in the amount of output and is therefore not affected by inflation. Furthermore, it is an operational indicator that statisticians collect on a monthly basis, which means that it very accurately reflects short-term trends. This is probably the first time that industry cycles have been assessed in such detail in regions within one country. As a rule, the authors previously limited themselves to assessing individual industry cycles without providing comparative estimates.

On the theoretical side, there is new evidence that border regions can develop much faster internally under current conditions, despite their remote geographical location. In this context, the policy of territorial development is focused on the most comprehensive use of the border location, in order to strengthen foreign economic ties. It is important that the dynamics of cycles are often proactive, and the shape of the curve, as it were, begins to hint at future changes.

The study has managerial implications since the methodology proposed by the authors allows comparing the dynamics between regions with a developed industry and those where an industry has significant potential that can be realized, by providing support at a proper time. Cyclical analysis provides an estimate of the time when industries in a region are most ready to realize the accumulated potential, and the cycle stage and the point of overshoot indicate the need for timely support (from regional authorities) to accelerate the development of the industry. The practical recommendations are that the cyclical dynamics of industries should be monitored, since cyclical analysis provides an estimate of the time when industries in the region are most ready to realize the accumulated potential, while the cycle stage and the minimum inflection point indicate the need for timely support (from authorities) to accelerate the development of an industry. In addition, cyclical analysis

indicates in advance the beginning of stagnation or a decline in the industry, which allows the business to respond in a timely manner.

The possibilities for increasing the amount of production are associated, first of all, with the improvement in territorial development policy and industrial policy and the application of a differentiated approach to its development and implementation, taking into account the cyclical nature of industries in certain regions.

The research implications are that it has identified new opportunities for a more in-depth study of patterns and factors of sectoral cyclicality, both in individual regions and in groups of regions that differ, for example, by the level of socio-economic development, geographical location, and resource potential, which can also be important for territorial development policy and industrial policy. We hope that these provisions provide guidance and recommendations for future research.

**Author Contributions:** Conceptualization, G.A.K. and V.K.S.; methodology, G.A.K. and V.K.S.; software, A.A.K.; validation, G.A.K., V.K.S., B.I.T., and A.A.K.; formal analysis, G.A.K.; investigation, G.A.K.; resources, A.A.K.; data curation, A.A.K.; writing—original draft preparation, G.A.K.; writing—review and editing, M.V.K., R.F., and B.I.T.; visualization, A.A.K.; project administration, G.A.K.; funding acquisition, G.A.K. All authors have read and agreed to the published version of the manuscript.

**Funding:** This research was funded by RFBR and FRLC, grant number 21-510-23002.

**Informed Consent Statement:** Not applicable.

**Data Availability Statement:** The calculations were carried out on the basis of publicly available data of the Unified Interdepartmental Information and Statistical System (UIISS) URL: https://www.fedstat.ru/ (accessed on 25 January 2023) (In Russian). Methodology, Rosstat, available online: https://rosstat.gov.ru/folder/14304 (accessed on 25 January 2023) (In Russian).

**Acknowledgments:** The authors also thank the journal editor and anonymous reviewers for their guidance and constructive suggestions.

**Conflicts of Interest:** The authors declare no conflict of interest.

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
