# Peer review of "Comparative Research of Internal and Border Regions: Analyzing the Differences in the Cyclical Dynamics of Industries for Industrial Policy and Territorial Development"

_economies, doi:10.3390/economies11030089_

Round 1

Reviewer 1 Report

The topic is rather timely, and its academic and policy relevance are important. The author shows some knowledge about spatial disciplines, nonetheless theoretical concepts are often used in a fuzzy way and are poorly explained too.

It is not entirely lucid, why the author has chosen four economic sectors for the analysis.

The time series look long enough, yet the period selected should be explained better.

More information on Russian (border) regions should be provided too. One can even imagine the short presentation of their typology.

Literature overview is insufficient and does not include some recent relevant sources that substantially widen the perspectives discussed in the paper, such as Suchacek (2022): Solid as a Rock: Media Portrayals of Cross-Border Activities, Sustainability 14(23), 15749 or Jakubowski et al (2022): Identifying Cross-Border Functional Area: Conceptual Backgorund and Empirical Findings from Polish Borderlands, European Planning Studies, 30:12, 2433-2455 and some others.

Hypothesis/hypotheses and RQ should be formulated clearly.

Influence of covid should be debated more.

One cannot find any relevant practical recommendations.

Author Response

We thank the reviewer for the careful reading, useful and clear comments that will help us to make the article better.

It is not entirely lucid, why the author has chosen four economic sectors for the analysis.

The following text is added (lines 90-95): For the analysis, the authors selected four sectors of the economy: the food industry, the petrochemical industry, the production of motor vehicles, trailers and semi-trailers, and the production of medicines and materials used for medical purposes. This choice is based on the availability of a relatively comprehensive statistical base for analysis, the level of the prevalence of these industries across Russian regions and their importance to the economy in the whole.

The time series look long enough, yet the period selected should be explained better.

The following text is added (lines 398-402): Data period for the research is January 2005 - December 2021, because it is from this period that official statistical bodies of Russia provide operational monthly data of the physical index of industrial production. This allowed the analysis to cover as much as possible cycles in the regions during this period and analyze the impact of global events (global crisis in 2008, sanctions against Russia in 2014).

More information on Russian (border) regions should be provided too. One can even imagine the short presentation of their typology.

The information is added in the form of a description (pp. 4-6) and Table 1.

Literature overview is insufficient and does not include some recent relevant sources that substantially widen the perspectives discussed in the paper, such as Suchacek (2022): Solid as a Rock: Media Portrayals of Cross-Border Activities, Sustainability 14(23), 15749 or Jakubowski et al (2022): Identifying Cross-Border Functional Area: Conceptual Background and Empirical Findings from Polish Borderlands, European Planning Studies, 30:12, 2433-2455 and some others.

The recommended sources (~ line 124 and line 150) and some other recent papers are added to the text (please see References).

Hypothesis/hypotheses and RQ should be formulated clearly.

The following text is added (lines 78-84): The article offers the following hypotheses:

H1: The industry cycle can be defined by an index of the volume of production.

H2: Despite the general economic conditions, the sectoral cycles vary significantly across regions and within groups of border and interior regions.

H3: In border regions, the cyclical volatility of industries is higher than in internal regions.

Influence of covid should be debated more.

We added Figure 4. Production of medicines and materials used for medical purposes in the Altai Krai (a) and Kaluga Oblast (b) and described for all analyzed industries the overall response to global shocks, such as the financial crisis of 2008-2011 and COVID-19.

However, there are global shocks to which individual industries have similar responses in many regions. The global financial crisis of 2008-2011 represented such a shock during the period under review. During the financial crisis, for example, there has been a cyclical decline in many regions of the chemical industry and in the production of motor vehicles, trailers and semi-trailers. During the COVID-19 period, in 2020 the regional situation was different, as can be seen in figures 1-4. We have also noticed an important point: the dynamic of the cycle is often preemptive before the influence of the big event begins. Thus, the financial crisis has exacerbated cyclical decline or stagnation in some regions that began before the acute phase of the financial crisis.     

It should be noted that the food industry and the production of medicinal products and materials used for medical purposes are, by their very nature, less sensitive to global shocks and, conversely, the industry cycle may shift to a growth phase during these periods.

We believe that the issue of the impact of global shocks on sectoral cycles requires more careful consideration. Anyway, thank you for the comment and an idea for a new article.

One cannot find any relevant practical recommendations.

The following text is added (lines 630-636): The practical recommendations are that the cyclical dynamics of industries should be monitored, since cyclical analysis provides an estimate of the time when industries in the region are most ready to realize the accumulated potential, while the cycle stage and the minimum inflection point will indicate the need for timely support (from authorities) to accelerate the development of an industry. In addition, cyclical analysis will indicate in advance the beginning of stagnation or decline in the industry, which will allow the business to respond in a timely manner.

Reviewer 2 Report

Dear Author(s),

The topic of the differences in cyclical dynamics of industries in internal and border regions is very interesting, and your manuscript, based on a good and novel methodological foundation, has the potential to be a valuable contribution to the discussion not only on cycles in the economy but also in research oriented towards the economic development of border areas. This does not mean, however, that the article does not have weaknesses to which I would like to draw attention. As the article introduces more novel content to the study of the economy of border regions rather than cycles in the economy (which is what I understand the authors' previous work referred to in the manuscript), my suggestions will be more in the area of border/regional studies.

Overarching remarks:

1.       The first and foremost comment relates to the concept of a region and, in particular, the concept of a border region. Bearing in mind the administrative structure of Russia and the sometimes huge area of some administrative units, the concept of a border region used in the article does not seem to be fully compatible with the commonly accepted perception of a border region in regional studies. This has far-reaching consequences. It is generally accepted that the backwardness of border areas is due to peripherality and so-called "border effects". Their negative impact is most substantial in the areas closest to the border. Therefore, even within the EU, there is criticism of an approach that considers relatively small NUTS3 units bordering other countries as border regions. Many studies consider border areas as those which are located at a maximum distance of 15 or 30 km from the border. The question arises - what is the impact of the border on industrial centres located several hundred kilometres from the national border? Therefore, the article would benefit from a more in-depth reflection on what role border location plays in the regional economy and why this impact can be expected to be negative.

2.       Staying in the same line, I found the paper lacked reference to the key concept of 'border effects' in this context. McCallum's (1995) study refers precisely to this phenomenon. It is worth recalling the term border effects and the evolution of the understanding of this phenomenon (starting from the classical international economics approach to border effects, looking at the trade flows reduction due to the presence of borders (which has been already done) and following with more recent approaches looking at the impact of borders on a more aggregate economic situation (Capello et al., 2018).

3.       I am aware that the last of my doubts is insurmountable. However, I wonder to what extent the comparison between border and interior regions in Russia is valid in the context of the conclusion that "the group of border regions in the food, chemical, and pharmaceutical industries is growing at a higher rate, contrary to a popular opinion about the backwardnesses of border regions". While I agree with the assertion that border regions have traditionally been perceived as backward, to what extent does this apply to Russia, where, as the authors point out, they account for some 47% of the regions in the country and 44% of the total population live in them? From this picture, it does not appear that the border regions are particularly underdeveloped in Russia, especially as they contain several metropolises that are locomotives of development, and many of them are located much closer to the core than some of the interior regions. It might be worth referring to some studies that would confirm the backwardness of the border regions in Russia. If this is not possible, the conclusions about the better performance of the border regions should not be generalised and applied only to the case of Russia.

Detailed remarks:

1.       The title of the article and the abstract - the title and the abstract do not indicate that the article is about the regions of Russia. This is significant and may be important for future readers primarily because the concept of regionalisation in Russia is not in line with the standards adopted in other countries. Thus border regions in Russia have slightly different characteristics. I think that this information should be included in the title or at least in the abstract.

2.       The introduction is based almost exclusively on Russian literature and is thus poorly integrated into the body of international literature. In general, the authors separate/distinguish the research results of Russian scientists in the text (p. 5, v 209). The literature review should be based on the subject criterion. I would suggest a slight rewording of the text.

3.       The sentence "In the Russian context, such border regions include border municipalities (Lazareva et al. 2020), borderlands (Svensson 2022) in EU policy and practice" (p. 3) is hard to understand.

4.       Paragraph 8 on page 5 - the sentence "This article uses software and methodological tools, described in detail in the book 244 (Semenychev et al. 2022)" and the reference to the publication is not clear. Please give a concise presentation of the tools used.

5.       P. 15, v. 520 - I firmly believe that in the sentence "in order to preserve geopolitical stability" the Author(s) are appealing for peace and good-neighbourly relations and not the imposition of the “Russkij mir” through military aggression on neighbouring states. If the Authors' intention is the latter, I would suggest deleting this sentence.

Author Response

We thank the reviewer for the careful reading, useful and clear comments that will help us to make the article better.

  1. The first and foremost comment relates to the concept of a region and, in particular, the concept of a border region. Bearing in mind the administrative structure of Russia and the sometimes huge area of some administrative units, the concept of a border region used in the article does not seem to be fully compatible with the commonly accepted perception of a border region in regional studies. This has far-reaching consequences. It is generally accepted that the backwardness of border areas is due to peripherality and so-called "border effects". Their negative impact is most substantial in the areas closest to the border. Therefore, even within the EU, there is criticism of an approach that considers relatively small NUTS3 units bordering other countries as border regions. Many studies consider border areas as those which are located at a maximum distance of 15 or 30 km from the border. The question arises - what is the impact of the border on industrial centres located several hundred kilometers from the national border? Therefore, the article would benefit from a more in-depth reflection on what role border location plays in the regional economy and why this impact can be expected to be negative.

Thank you for your valuable comment. Indeed, The territorial organization in Russia implies the division into municipal entities and their association into larger areas - regions (Russian subjects) that differ significantly in scale. For example, the distance from the regional center of one of Russian border regions - the city of Novosibirsk - to the border with Kazakhstan is 480 km. Therefore, Russia has already left from the «narrow» understanding of the border territory for the purposes of regional development, understanding that the reduction of negative «border effects» is only possible to consider border municipalities on a small border strip as an integral part of the economy of the region as a whole. The location of borders plays an important role in the regional economy, regardless of the distance to the industrial center of the region. If this distance is more than 100 km, the impact of the peripheral effect - remoteness from the centre of the region, where the main centres of education, health and other services tend to be concentrated - increases. In this sense, the border regions or regions with border municipalities in their composition in Russia are in a more difficult situation than the interior regions and depend to a large extent on how simple trade and movement regimes are with neighbouring territories.

It is no coincidence that in documents of strategic character the border regions are identified as a separate category (Russian Federation 2020). Thus, in the Strategy for Spatial Development of the Russian Federation until 2025 (Russian Federation 2019), 21 subjects of the Russian Federation, located along the land border of the country, are divided into 4 groups depending on the contiguity with the state being a member of an international association of countries. The composition of these groups and certain administrative, territorial and socio-economic characteristics are presented in Table 1.

Although balanced polycentric development is still a big problem for Russia, there are examples of successful border regions. For example, the ten most successful regions in terms of economic development in 2022 included the border regions of the Samara region and the Tyumen region, which occupy 9 and 10 places respectively. However, given that there are more than 20 border geostrategic territories in Russia - this is not so much.

  1. Staying in the same line, I found the paper lacked reference to the key concept of 'border effects' in this context. McCallum's (1995) study refers precisely to this phenomenon. It is worth recalling the term border effects and the evolution of the understanding of this phenomenon (starting from the classical international economics approach to border effects, looking at the trade flows reduction due to the presence of borders (which has been already done) and following with more recent approaches looking at the impact of borders on a more aggregate economic situation (Capello et al., 2018).

The article has a reference to the paper (McCallum 1995) McCallum, John. 1995. National borders matter: Canada–US regional trade patterns. The American Economic Review, 85(3) : 615–623, http://www.jstor.org/stable/2118191.

One more reference is added: (Capello at al. 2018) Capello, Roberta, Andrea Caragliu, and Ugo Fratesi. 2018. "Measuring border effects in European cross-border regions." Regional Studies 52.7 : 986-996.

  1. I am aware that the last of my doubts is insurmountable. However, I wonder to what extent the comparison between border and interior regions in Russia is valid in the context of the conclusion that "the group of border regions in the food, chemical, and pharmaceutical industries is growing at a higher rate, contrary to a popular opinion about the backwardnesses of border regions". While I agree with the assertion that border regions have traditionally been perceived as backward, to what extent does this apply to Russia, where, as the authors point out, they account for some 47% of the regions in the country and 44% of the total population live in them? From this picture, it does not appear that the border regions are particularly underdeveloped in Russia, especially as they contain several metropolises that are locomotives of development, and many of them are located much closer to the core than some of the interior regions. It might be worth referring to some studies that would confirm the backwardness of the border regions in Russia. If this is not possible, the conclusions about the better performance of the border regions should not be generalised and applied only to the case of Russia.

We have removed from the text the phrase from the abstract "contrary to the common opinion about backwardness of border regions". Indeed, not everything is so clear about Russia and there are examples of successful border regions. For example, the ten most successful regions in terms of economic development in 2022 included the border regions of the Samara Oblast and the Tyumen Oblast, which occupy 9 and 10 places respectively. However, given that there are more than 20 border geostrategic territories in Russia - this is not so much.

Detailed remarks:

  1. The title of the article and the abstract - the title and the abstract do not indicate that the article is about the regions of Russia. This is significant and may be important for future readers primarily because the concept of regionalisation in Russia is not in line with the standards adopted in other countries. Thus border regions in Russia have slightly different characteristics. I think that this information should be included in the title or at least in the abstract.

Thank you for the comment, the text of the abstract was changed, links to the Russian context of the study were added. In addition, a text and a corresponding table (Table 1) containing a typology of Russian border regions was added to the text of the article.

  1. The introduction is based almost exclusively on Russian literature and is thus poorly integrated into the body of international literature. In general, the authors separate/distinguish the research results of Russian scientists in the text (p. 5, v 209). The literature review should be based on the subject criterion. I would suggest a slight rewording of the text.

Thanks for the comment, the text of the introduction was changed, references to the works of foreign authors were added.

  1. The sentence "In the Russian context, such border regions include border municipalities (Lazareva et al. 2020), borderlands (Svensson 2022) in EU policy and practice" (p. 3) is hard to understand.

Thank you for the comment, this sentence was changed for a better understanding.

  1. Paragraph 8 on page 5 - the sentence "This article uses software and methodological tools, described in detail in the book 244 (Semenychev et al. 2022)" and the reference to the publication is not clear. Please give a concise presentation of the tools used.

This book contains a detailed description of those tools and methodologies that are summarized in the article in the entire section 3. Methodology. We added a clarification on the composition of the toolkit.

  1. P. 15, v. 520 - I firmly believe that in the sentence "in order to preserve geopolitical stability" the Author(s) are appealing for peace and good-neighbourly relations and not the imposition of the “Russkij mir” through military aggression on neighbouring states. If the Authors' intention is the latter, I would suggest deleting this sentence.

We were surprised to hear that, but we deleted the phrase.

Round 2

Reviewer 2 Report

Dear Authors,

Thank you for your amendments. I am satisfied with the changes to the article, which have certainly raised its quality and cleared up a few doubts. Before publishing the article, I would ask you to correct two minor issues:

- v. 38 - "border effects", not "boundary effects" (this is not the same)

- p. 5, v. 210 - I like the table you have added to the paper, hovewer I have some doubts about it. The first refers to the Kalinngrad oblast - is it not considered a border region? The second one applies to the Republic of Abkhazia which is not recognised as a state under international law (only Russia, Nauru, Nicaragua, Syria and Venezuela recognise it). I realise that this is problematic so I suggest using a solution that is often used in international literature (Republic of Abkhazia/Georgia) or adding a reference next to the table (not only in the text) that would point directly to the source of such information.  

Author Response

Dear Reviewer,

Thank you very much for your careful reading and indication of errors detected. We have corrected “border effects”. We also have added the explanation of why the Kaliningrad Oblast is not found in Table 1 (though we have included it into the analysis of regional cycles). The bordering state Republic of Abkhazia is also changed into Republic of Abkhazia / Georgia – thank you again for this variant.